# Porous Structural Properties of K or Na-Co Hexacyanoferrates as Efficient Materials for CO_2_ Capture

**DOI:** 10.3390/ma16020608

**Published:** 2023-01-08

**Authors:** Paloma M. Frías-Ureña, Maximiliano Bárcena-Soto, Eulogio Orozco-Guareño, Alberto Gutiérrez-Becerra, Josué D. Mota-Morales, Karina Chavez, Víctor Soto, José A. Rivera-Mayorga, José I. Escalante-Vazquez, Sergio Gómez-Salazar

**Affiliations:** 1Departamento de Química, Universidad de Guadalajara (CUCEI), Boulevard Marcelino García Barragán #1421, Esquina Calzada Olímpica, Guadalajara 44430, Mexico; 2Departamento de Ciencias Básicas y Aplicadas, Universidad de Guadalajara (CUTonala), Avenue Nuevo Periférico 555, Tonalá 45425, Mexico; 3Centro de Física Aplicada y Tecnología Avanzada, Universidad Nacional Autónoma de Mexico, Querétaro 76230, Mexico; 4Graduate Program in Materials Science, Departamento de Ingeniería de Proyectos, Universidad de Guadalajara (CUCEI), Boulevard Marcelino García Barragán #1421, Esquina Calzada Olímpica, Guadalajara 44430, Mexico; 5Departamento de Ingeniería Química, Universidad de Guadalajara (CUCEI), Boulevard Marcelino García Barragán #1421, Esquina Calzada Olímpica, Guadalajara 44430, Mexico

**Keywords:** hexacyanoferrates, porosity, adsorption, CO_2_ capture, isosteric heat of adsorption

## Abstract

The stoichiometry of the components of hexacyanoferrate materials affecting their final porosity properties and applications in CO_2_ capture is an issue that is rarely studied. In this work, the effect that stoichiometry of all element components and oxidation states of transition metals has on the structures of mesoporous K or Na-cobalt hexacyanoferrates (CoHCFs) and CO_2_ removal is reported. A series of CoHCFs model systems are synthesized using the co-precipitation method with varying amounts of Co ions. CoHCFs are characterized by N_2_ adsorption, TGA, FTIR-ATR, XRD, and XPS. N_2_ adsorption results reveal a more developed external surface area (72.69–172.18 m^2^/g) generated in samples containing mixtures of K^+^/Fe^2+^/Fe^3+^ ions (system III) compared to samples with Na^+^/Fe^2+^ ions (systems I, II). TGA results show that the porous structure of CoHCFs is affected by Fe and Co ions oxidation states, the number of water molecules, and alkali ions. The formation of two crystalline cells (FCC and triclinic) is confirmed by XRD results. Fe and Co oxidation states are authenticated by XPS and allow for the confirmation of charges involved in the stabilization of CoCHFs. CO_2_ removal capacities (3.04 mmol/g) are comparable with other materials reported. CO_2_ adsorption kinetics is fast (3–6 s), making CoHCFs attractive for continuous operations. Q_st_ (24.3 kJ/mol) reveals a physical adsorption process. Regeneration effectiveness for adsorption/desorption cycles indicates ~1.6% loss and selectivity (~47) for gas mixtures (CO_2_:N_2_ = 15:85). The results of this study demonstrate that the CoHCFs have practical implications in the potential use of CO_2_ capture and flue gas separations.

## 1. Introduction

The need to lessen climate change produced by greenhouse gases, such as CO_2_, is a reality nowadays. To this end, a great variety of methodologies have been applied in the removal of CO_2_. Amongst these, the use of porous adsorbents displaying fast removal velocities, improved uptake capacities, increased affinities for CO_2_ in the presence of other gases, and stable regeneration properties have shown promise as an alternative to more conventional methods in the solution to this problem and are critical when selecting the adsorbent [1]. Particularly, the use of Metal-Organic Frameworks (MOFs) possessing highly developed porosities and outstanding textural properties, e.g., high specific surface areas, optimum pore diameters to ease CO_2_ diffusion, and tunable chemical and structural properties have appeared as an encouraging alternative [2,3]. For instance, very recently, MOF-74(Ni)-24-140 was reported to present an adsorption capacity of CO_2_ of 8.29/6.61 mmol/g at 273/298 K under the normal pressure of 1.0 bar [4], and this exceeded those of materials such as the amine-functionalized carbon nanotubes TEPA IG-MWCNT of 3.09 mmol g^−1^ [5] and MOF-808 functionalized with ethyleneamines (~2.5 mmol g^−1^) [6]. Before this, a type of MOFs, metal hexacyanoferrate (also known as Prussian Blue Analogs (PBAs)), attracted increasing interest in CO_2_ removal applications due to their intricate micro/meso porous structures, highly developed textural properties and the relatively mild and easy to implement synthesis routes [7]. PBAs have been used in CO_2_ capture or H_2_ storage applications due to their highly porous properties [8,9]. A recent study demonstrated that hexacyanoferrates, e.g., K_2x/3_Cu^II^[Fe^II^_x_Fe^III^_1-x_(CN)_6_]_2/3_, containing alkali ions such as K^+^ comply with these requirements [10]. These materials have demonstrated that they possess outstanding CO_2_ capture capacities (~3.0 mmol/g CO_2_), fast adsorption kinetics, high selectivity indices for gas mixtures such as CO_2_/N_2_, and highly stable regeneration capacities [1].

PBAs are coordination compounds with a general formula M^+^_p_M’_q_[M’’(CN)_6_]_r_·xH_2_O where M’ and M’’ are commonly transition metals that present different oxidation states. Synthesis conditions and the presence of alkali M^+^ ions have an influence on the stoichiometric coefficients p, q, r, and x and the structural and porous properties. These coordination compounds possess a crystalline structure [11]. PBAs have novel applications due to their electronic, magnetic, optical, electrochromic, electrocatalytic, ion sensing, and ion storage properties [12]. In this sense, cobalt hexacyanoferrates (CoHCFs) have been extensively studied due to their physicochemical, electrochromism properties, photoinduced reversible magnetization, and reversible thermochromism [13], properties which derive from the bivalent and trivalent oxidation states of iron and cobalt of the compound [14], so they have been used to manufacture modified graphite electrodes [15]. Furthermore, these materials have also been used in technologies applied in the decontamination of radioactive cesium [16]. CoHCFs have a crystalline structure containing ferricyanide vacancies occupied by water molecules [17,18]. Water molecules form hydrogen bridge networks at the interstitial places where alkali ions are also inserted. The stoichiometry of the compound can vary depending on the number of alkali ions present, and network electroneutrality is guaranteed via the adjustment of [Fe^III^(CN)_6_]^3−^ gaps in the structure [19]. When water molecules are removed from the structure, the material becomes porous with the capability of capturing gases such as CO_2_ [10]. There are two types of positions where water molecules can be located within the crystalline structure of the CoHCFs, namely, water coordinated to Co ions at vacant sites of the [Fe^III^(CN)_6_]^3−^, and water contained inside every eighth of the crystal unit cell known as zeolitic water [20]. Consequently, different types of pores are present in the CoHCFs analyzed, where the formation of micropores has been attributed to the removal of zeolitic and coordination water, whereas the formation of mesopores has been attributed to the agglomeration of particles [20,21]. The gaps can be interconnected, forming micro tunnels [22], thus improving the textural properties and increasing their porosity.

The effect of porous and textural properties of PBAs on the extent of CO_2_ capture has been studied. There are few studies about the effect of incorporated alkali ions on the porous structure of PBAs. For instance, Li et al. [23] recently reported that K^+^ ions exposed after water molecules removal from pores/crystals of PBAs produced adsorption sites owing a high affinity for CO_2_ molecules with the consequent modification of the porous structure of the material. On the other hand, Karadas et al. [20] reported that available free pore volumes, V_free_ calculated from Monte Carlo simulations on PBAs samples, enabled the adsorption of significant amounts of CO_2_ on the PBAs structures. In addition, they also reported that the adsorption of CO_2_ on large pores resulting in random gaps would be more favorable, especially at low and moderated pressures. In this regard, there is controversy in the reports that indicate, on the one hand, that adsorption of CO_2_ or H_2_ is preferential on large pores of PBAs [24], whereas on the other hand, based upon infrared spectroscopic studies on M_3_[Co(CN)_6_]_2_, Natesakhawat et al. [25] reported that the adsorption of CO_2_ on these materials was unaffected by the differences between large and small pores. In a recently published review, Lopez et al. [26] reported that the formation of open cavities on the 3D frameworks (such as those present in PBAs) supports ion or gas molecule diffusion. These structural properties, combined with high porosity, clear the way for their use in gas capture and storage applications. Regardless, the above statements indicate the importance of studies dealing with the porous structural properties of PBAs.

The aim of this work is to report on the effect that synthesis stoichiometry has on the structural and porous properties of K or Na-CoHCFs and the efficient CO_2_ capture. The novelty of this work includes reporting on the effect that stoichiometric amounts, the types of alkali K and Na ions, and the oxidation state of transition metals have on the final structural and porous properties of CoHCFs and the CO_2_ adsorption capacities. A battery of instrumental techniques is applied for the characterization of CoHCFs, including N_2_ adsorption, TGA, FTIR-ATR, XRD, and XPS.

## 2. Experimental

### 2.1. Synthesis of K or Na-Cobalt Hexacyanoferrates (CoHCFs)

The synthesis of CoHCFs was performed using the drop method and a volumetric ratio of K_3,4_/Na_4_Fe(CN)_6_:Co(NO_3_)_2_ of 6:1. In a typical synthesis, 300 mL of a solution 0.02 M of [Fe(CN)_6_ ]^n−^ was placed in an Erlenmeyer flask under stirring. To this, 50 mL of a solution of Co(NO_3_)_2_ at varying concentrations of 0.035–0.175 M was added dropwise at time intervals of 2–3 s between each drop addition at 25 °C. After the addition of all the nitrate solution, the reacting mixture was agitated for another 5 min and then was left to rest for 24 h to allow the system to stabilize. All CoHCFs solutions were stored in amber bottles to avoid the reduction of the transition metals by sunlight. To obtain CoHCFs powders, solutions were centrifuged for 20 min, the supernatant removed, and the resulting precipitate washed with doubly distilled water. The precipitate was then dried at 60 °C for 48 h. The powder obtained was ground using an Agatha mortar. Stoichiometric coefficients were determined from ICP-MS after acid digestion. The atomic contents were normalized against Fe. Alkali metals were obtained by charge balances. The H_2_O content was determined from TGA. Sample identifications are shown in Table 1.

### 2.2. K or Na-CoHCFs Characterization

#### 2.2.1. Chemical Analyses

The contents of iron, cobalt, sodium, and potassium of the synthesized CoHCFs were obtained by ICP-MS (Agilent Technologies, 7500a model) of the digested samples. Digestion was conducted with boiling sulfuric acid. The contents of C and N were obtained by combustion at 1100 °C using a Leco TruSpec Micro CHNS elemental analyzer (St. Joseph, MI, USA).

#### 2.2.2. N_2_ Adsorption Measurements

Nitrogen adsorption measurements were conducted using an ASAP 2020 Micromeritics Sorptometer (Norcross, GA, USA) at 77 K on CoHCFs to study their textural properties and the effect of varying amounts of water on the specific surface areas and pore sizes. About 0.3 g of the sample was conditioned under a heat flow at 100 °C for 6 h under vacuum (10 µm Hg) to remove all impurities (e.g., CO_2_ and H_2_O) previously adsorbed onto the surface of samples that otherwise cause an error in the surface area measurement. The Brunauer-Emmet-Teller (BET) specific surface area was calculated by the BET equation using adsorption data within the domain of the BET equation in the relative pressure range of 0.05 < P/P_o_ < 0.26; the total pore volume (V_t_) was determined by nitrogen adsorption at P/P_o_ = 0.995; the pore size (D_p_), and pore size distribution was obtained from the desorption branch of the isotherm measurements by the BJH method. The t-plot method was used to estimate the micropore volume (V_mic_), external surface area (S_ext_), and micropore area (S_mic_) using the Harkins and Jura equation in the thickness range of 0.35–0.50 nm. Mesopore volume (V_meso_) was calculated as the difference between the total volume of pores (V_t_) and micropore volume (V_mic_).

#### 2.2.3. Thermogravimetric Analysis (TGA)

Thermogravimetric measurements (TGA) were conducted on CoHCF to study the %wt of water loss as a function of temperature. TGA analyses were performed using a Discovery model analyzer from TA Instruments (Amarillo, TX, USA). The analyses were performed using dried samples. About 10–15 mg of sample was placed in a platinum pan; a temperature scanning in the range of 25–600 °C was used, but a shorter temperature range (25–500°C) was shown in the thermograms to indicate more clearly the weight loss due to water molecules; the heating rate was 10 °C min^−1^ under a nitrogen atmosphere at 60 mL min^−1^.

#### 2.2.4. FTIR-ATR

Fourier Transformed Infra-Red Attenuated Total Reflection spectra (FTIR-ATR) of the CoHCFs were determined with a Bruker Alpha FTIR-ATR system (Bruker Optics, Billerica, MA, USA) utilizing a 300 Golden Gate diamond ATR Model. Samples were scanned over the range of 4000–450 cm^−1^ at a resolution of 2 cm^−1^.

#### 2.2.5. X-ray Powder Diffraction Data Collection and Analysis

A Rigaku diffractometer model Ultima IV was used with CuKα radiation at λ = 1.54051, a 2Ɵ range of 4°–90°, a step size of 0.026, a time step of 30 s, an intensity of 30 mA, and a power of 40 kV. The Rietveld method implemented in the program MAUD (Materials Analysis Using Diffraction) [27] was used for structure analysis and refinement of sample M6, whereas samples M1 and M11 were refined peak by peak.

#### 2.2.6. XPS Analysis

X-ray Photoelectron Spectroscopy (XPS) measurements were conducted on selected samples. XPS data were collected with an XR50 M monochromatic Al Kα (hν = 1486.7 eV) X-ray source, and a Phoibos 150 spectrometer with the one-dimensional detector 1DDLD provided by SPECS (Berlin, Germany) operated at a potential of 10–14 kV. The base pressure in the analyzer was less than 5.1 × 10^−9^ Torr. XPS spectra were taken using an aluminum anode X-ray source, operating at 250 kW (12.5 kV and 20 mA). High-resolution scans were acquired over the range of 50–70 eV (Co and Fe 2p) with the pass energy adjusted to 15 eV and step 0.1 eV wide. After baseline subtraction, the spectra were deconvoluted using peak fitting analysis with the AAnalyzer software, and peak positions were found. Peak assignments were determined from literature reports.

#### 2.2.7. Surface Morphology of CoCHFs

The surface morphology of CoHCFs was studied by Scanning Electron Microscopy (SEM) using a Tescan Mira3 LMU high-resolution microscope operated at 15 kV with the detector operating at 13.3–13.4 mm distance and Transmission Electron Microscopy (TEM) using a JEOL JEM 1010 100 kV microscope.

#### 2.2.8. CO_2_ Adsorption on CoHCFs

CO_2_ adsorption and desorption isotherms on CoHCFs were obtained using an ASAP 2020 Micromeritics Sorptometer (Norcross, GA, USA) equipped with an ISO Controller Micromeritics unit for temperature control with the stability of ±0.1 °C. The analysis gas was supplied by INFRA (México City, México) with 99.995% purity. Isotherms were obtained in the temperature range of 273–353 K in the relative pressure interval of 0.00–1.00 P/Po (0–100 kPa). Prior to the measurements, about 80–100 mg of previously dried samples (60 °C, 72 h) were degassed at 150 °C under vacuum for 6 h. At this temperature, only the loss of water is observed, and there is no decomposition of the CN group of the CoHCFs [28]. An equilibration time of 5 s was used since higher times produced excessive acquisition data of the isotherms. The rate of adsorption (ROA) of CO_2_ on CoHCFs was measured at three temperatures (273, 298, and 313 K) by recording pressure change per dose vs. time (or measuring the time it took a dose of gas to adsorb) with the integrated ROA function in the ASAP 2020 Micromeritics Sorptometer.

#### 2.2.9. Isosteric Heats of Adsorption, Q_st_

The isosteric heats of adsorption were calculated with the incorporated function in the MicroActive software of the ASAP 2020. Q_st_ at each adsorption point was assessed using the Clausius-Clapeyron equation [29] in the form:(1)$$Q_{st}=-R{\left(\frac{\partial \ln P}{\partial \left(\frac{1}{T}\right)}\right)}_{\theta }$$
where *R* is universal gas constant, *T* is the analysis temperature (K), *P* is the partial pressure of CO_2_, and *θ* is the adsorption quantity or extent to which the sample surface is covered with adsorbate.

#### 2.2.10. Stability and Regenerability of CoHCFs

The stability and regenerability of CoHCFs after CO_2_ adsorption/desorption cycles were investigated. Five runs were performed. In the first cycle, samples were degassed for 6 h at 150 °C under vacuum; for the rest of the cycles, samples were conditioned under vacuum at the same temperature but for 2.5 h to avoid the collapse of the crystalline structure of samples. All cycles were run at 298 K.

## 3. Results and Discussion

### 3.1. N_2_ Adsorption and Textural Properties of CoCHFs

All samples present type IVa isotherms and hysteresis loop type H1 according to the IUPAC classification [30], Figure 1a–c, characteristic of mesoporous materials, except samples M9 and M10, which are type II with H3 type hysteresis loop. In the case of type H1 hysteresis loop, this is characteristic of solids that present narrow ranges of invariable mesopores; the H3 type hysteresis loop is characteristic of non-rigid aggregates of plate-like particles and pore networks comprised of macropores [30]. On the other hand, type II isotherms present characteristics of physisorption of most gases on nonporous adsorbents. In this case, unlimited combined monolayer and multilayer adsorption takes place, thus conferring the shape of this isotherm [31]. Adsorption/desorption isotherms reveal that samples containing K^+^ ions and a mixture of Fe^2+^/Fe^3+^ ions (see Section 3.3 and Section 3.5) of system III exhibit a porous structure with a much more developed external surface area, S_ext_, (72.69–172.18 m^2^ g^−1^) and a total pore volume V_t_, (0.215–0.505 cm^3^ g^−1^) more uniform as compared to samples containing Na^+^ and Fe^2+^ ions (systems I and II) (S_ext_ of system I is 49.15–105.88 m^2^ g^−1^, S_ext_ of system II is 0.4720–91.65 m^2^ g^−1^, V_t_ of system I ~ 0.061–0.444 cm^3^ g^−1^ and V_t_ of system II is 0.001–0.228 cm^3^ g^−1^). In the case of the samples of system II, textural parameters are too small compared to the samples of the other two systems. For instance, samples M9 and M10 reveal an S_BET_ of 5.72 and 0.45 m^2^ g^−1^, respectively. These values indicate that these samples are nonporous materials, as can be observed from the corresponding pore size distributions (Figure 1e).

The specific surface areas decrease as the Na_4_Fe(CN)_6_/Co(NO_3_)_2_ and K_3_Fe(CN)_6_/Co(NO_3_)_2_ molar ratios decrease for samples of systems I and III but not for samples of system II, see Table 1. The presence of Na ions, the +2-oxidation state of Fe in system I, and the presence of K^+^ ions and both Fe^2+^/Fe^3+^ oxidation states in system III produce CoHCFs with stoichiometries in which the number of alkali ions and water molecules depend on the reactant initial molar ratios. This behavior can be explained by considering the final number of Co ions in the CoHCFs. In these systems, the number of Co ions in the final stoichiometry is higher (or similar) than the number of Fe ions; thus, it is likely to find Co^2+^ ions not coordinated to C≡N groups, promoting the formation of cubic cells with high specific surface area. This generates a higher number of pores [32], favoring and contributing to the specific surface areas of samples, as is more noticeable in the case of samples M11-M15. Ferro/ferri-cyanide vacancies are filled by coordination water, which, when removed by the heat treatment (100 °C) during the conditioning process, leaves empty spaces generating internal pores inside the material that are joined, forming interconnected tunnels conforming the porous structure of the CoHCFs, as reported by Ojwang et al. [10]. On the other hand, samples M1–M5 depict a range of surface areas S_BET_ of 52.7–311.9 m^2^ g^−1^. This behavior is not observed in system II. It is worth mentioning that even though the only difference between systems I and II is the alkali ion, in the latter, the final stoichiometry shows a lower number of Co ions, promoting a decrease in the specific surface area, which is confirmed by the decrease in the water molecules compared to the other systems. In the case of samples M9 and M10, their specific surface areas are too low (5.72 and 0.45 m^2^ g^−1^, respectively) to fall within the limit of detection of the instrument. These results confirm the impact that stoichiometry has on the final porous properties of the CoHCFs. In the case of the samples of systems I and III, an FCC crystal is obtained (see Section 3.4), and these samples are the ones that present the highest amounts of N_2_ adsorbed, whereas the samples of system II present a triclinic cell and the lowest amount of nitrogen adsorbed. These differences can be due to the fact that the formation of porosity in the FCC system is facilitated by the geometry of this system, while in the case of the triclinic system, this geometry prevents the formation of the interconnected pores. Furthermore, when Co concentration is varied during synthesis, the crystalline structure remains, but textural properties, such as porosity, change. The crystalline arrangement obtained in samples of each system is independent of the Co concentration used, given that all samples were synthesized under the same conditions of pressure and temperature. In the case of pore volume, V_t_, data from Table 1 reveal that the values for samples within system I are higher in samples synthesized with higher Co(NO_3_)_2_ concentration and present the trend M1 > M2… > M5; this trend is also observed in the case of samples of system III, with M11 > M12…. > M15. However, as in the case of S_BET_, samples of system II do not show this trend. On the other hand, in the case of D_P_, the trend observed is variable for the three systems. In the case of pore size distributions, Figure 1d–f, the distributions observed are very narrow (except for samples M1 and M10) and present a monomodal shape in the mesopore region of 2.5–10.0 nm attributed to the hysteresis loop present and suggesting the formation of a structure with more uniform pores. The uniformity of pore diameters is confirmed by the steep slope of the desorption part of the isotherm. The results obtained from N_2_ adsorption measurements provide evidence of the mesoporosity obtained during the synthesis procedure through the formation of interconnected tunnels and cavities formed in the crystalline structure of CoHCFs. The N_2_ adsorption measurements present large pore sizes of about 30 to 300 Å. These can be the entrances of interconnected channels through which CO_2_ molecules are transported to the smaller pores. These large pores are presumably formed by agglomerated particles and not by part of the crystal structures, which have two intrinsic types of cavities, ones ~5 Å wide formed by the framework where alkali ions and H_2_O molecules may reside, and the other ones ~10 Å wide formed by absent [Fe(CN)_6_]^n−^ groups, according to Svensson et al. [8]. CO_2_ is adsorbed into these cavities. These results suggest that the oxidation states of Co and Fe, the nature of alkali ions, and the water content modify the way particles of CoHCF agglomerate. Consequently, when some of these variables are modified, the adsorption properties of each family change. From the formulas shown in Table 1, it can be observed that the amount of alkali ions increases as the concentration of Co decreases, and this is due to the need of the crystals to satisfy the charge balances.

### 3.2. TGA Analysis

TGA measurements were performed to determine the amount of water contained in CoHCFs and to see any changes in the chemical structure of the samples when subjected to thermal decomposition. In all cases, a temperature range of 25 to 500 °C was used to ensure that all the water molecules were removed. The thermograms are shown in Figure 2. The weight loss at low temperatures, around 25–150 °C, corresponds to the physically adsorbed or zeolitic water [28]. At ~150–250 °C, the weight loss corresponds to the lattice or coordinated water [28]. The decomposition of the samples with the liberation of C≡N units is carried out at 250–500 °C [33]. The weight loss of water for each sample was obtained from the thermograms using the derivatives of the data (Appendix A). The data were used to determine the corresponding number of water molecules in the formulas of CoHCFs, as shown in Table 1. Although the molar ratio of the reactants varied in the same way in the three systems (Table 1), the obtained stoichiometries do not show the same trend. This behavior is common in CoHCFs, where even the way of mixing the reactants (e.g., dropwise or immediate mixing) has a remarkable effect on the final stoichiometry [14]. Thus, we expect to obtain CoHCFs with diverse stoichiometries due to the reactant excesses used in this work. Nevertheless, it is possible to observe a general behavior between the systems. For instance, the number of water molecules strongly depends on the number of alkali ions in the structure. This behavior is more prominent in systems I and III, where the number of water molecules decreases as the number of alkali ions increases. This behavior suggests that the introduction of more alkali ions into the CoHCFs framework reduces the amount of coordinating water in the unit cell [34]. This behavior is also followed in system II in which the number of alkali ions in the CoHCFs is greater than in the other systems, and it is expected that the number of water molecules is also less than in the other systems, as is shown in Table 1. The number of water molecules in system I and especially in system III may be the reason for the significative textural properties of these samples. On the other hand, the poor textural properties of system II can be explained by the small number of water molecules in the CoHCFs framework.

### 3.3. FTIR-ATR Study

FTIR is commonly used to characterize CoHCFs [13,14,35,36,37,38]. Figure 3a–c shows the FTIR spectra for all the samples. The position of the band due to the CN group in the range of 2200 to 2000 cm^−1^ (shown in spectra of three systems) is useful to qualitatively determine the oxidation states of the metallic ions conforming to the CoHCFs, according to literature reports [13,14,35,36,37,38]. On the other hand, bands in the range of 3650 to 3100 cm^−1^ and 1600 to 1630 cm^−1^ indicate the presence of water molecules [35,36,37,38]. As expected, the spectra of Figure 3a,b are similar because systems I and II samples were prepared with [Fe(CN)_6_]^4−^ ions. In both cases, the presence of a single band at ca. 2077 cm^−1^ indicates that the structure of the CoHCFs is based on the Fe^2+^–CN–Co^2+^ chain, regardless of the concentration ratio [X_4_[Fe(CN)_6_]]/[Co(NO_3_)_2_] (where X= Na or K) used in the synthesis of CoHCFs. This is confirmed by the presence of bands at ca. 590 and 448 cm^−1^ (samples M1–M5 and M6–M10). The presence of Co^3+^ in systems I and II samples could not be detected by FTIR, whereas the presence of this species was detected by XPS in the samples of the three systems (Section 3.5). The main differences between the spectra of these samples are the intensities of the bands due to the presence of water molecules. It has been proposed [35,36] that bands in the range of 3650–3100 cm^−1^ are assigned to the presence of adsorbed water molecules, while bands in the range of 1600–1630 cm^−1^ indicate the presence of water molecules that are trapped within the crystal structure (either coordinated or zeolitic). According to the intensity of these bands, it is evident that CoHCFs prepared with Na_4_[Fe(CN)_6_] (system I) contain more water molecules than the ones prepared with K_4_[Fe(CN)_6_] (system II) for all the concentration ratios used, which confirms the results obtained from TGA.

Unlike the previous cases, Figure 3c (spectra of samples of system III) presents an extraordinary difference in the bands in the range of 2200 to 2000 cm^−1^ as a function of the concentration ratios [K_3_[Fe(CN)_6_]/[Co(NO_3_)_2_]. At the highest Co(NO_3_)_2_ concentration (sample M11), the presence of two bands at ca. 2160 and 2110 cm^−1^ is evident, which indicates that the CoHCFs synthesized at these conditions are formed by the mixture of two chains Fe^3+^–CN–Co^2+^ and Fe^2+^–CN–Co^3+^, respectively. The presence of Fe^2+^/Fe^3+^ and Co^2+^/Co^3+^ ions is also confirmed by the XPS results. The appearance of the Fe^2+^–CN–Co^3+^ chain is due to the spontaneous electron transport between Fe^3+^ and Co^2+^ ions according to the reaction Fe^3+^–CN–Co^2+^ → Fe^2+^–CN–Co^3+^ [13,14]. The mixture of chains is also observed by the bands at 544 and 425 cm^−1^ and a small band at 590 cm^−1^ [39]. For CoHCFs prepared at lower Co(NO_3_)_2_ concentrations, the relative intensity of the band assigned to the Fe^3+^–CN–Co^2+^ chain decreases, while the intensity of the band assigned to the Fe^2+^–CN–Co^3+^ chain increases. This behavior suggests that when fewer Co ions are available, the electron transport inside the structure of the CoHCFs increases. This can be explained by considering that both chains, Fe^3+^–CN–Co^2+^ and Fe^2+^–CN–Co^3+^, can be expressed as Co^2+^_3_[Fe^3+^(CN)_6_]_2_ and Co^3+^_4_[Fe^2+^(CN)_6_]_3_, respectively. In the former, 33% of the Co^2+^ ions are not coordinated with the C≡N group, while in the latter, this percentage decreases to 25%. Consequently, when the concentration of Co^2+^ ions decreases, the formation of the chain Fe^2+^–CN–Co^3+^ is favored [13,14,37,38]. This can also be confirmed by the relative intensity of the bands due to the presence of water molecules. To maintain electroneutrality, the Co ions not coordinated to the C≡N group are coordinated with water molecules. Then, for the Fe^3+^–CN–Co^2+^ chain, where there are more Co^2+^ ions available, more water molecules will be in the CoHCFs structure [13,14,37,38]. When the concentration of the Fe^2+^–CN–Co^3+^ chain increases, there is a decrease in the water molecules needed to stabilize the crystal. Thus, a decrease in the relative intensity of the bands due to the presence of water is observed as the band also decreases due to the chain Fe^3+^–CN–Co^2+^.

### 3.4. XRD Analysis

The crystalline structure of the as-prepared CoHCFs was determined by XRD (Figure 4). Crystallographic data and structural parameters obtained after refinement are presented in Table 2 for samples M1, M6, and M11 and Appendix A. For the analysis of crystalline structure, similar results are obtained for samples within each system; because of this, the results of diffractograms corresponding to the first sample of each system are presented for representativeness (i.e., samples M1, M6, and M11, Figure 4a). It can be observed that either the Fe oxidation state or the type of alkali ion (Na^+^ or K^+^) influences the formation of the different types of structures. Sample M1 has Na^+^ and Fe^2+^ ions; sample M6 has K^+^ and Fe^2+^ ions; and sample M11 has K^+^ and a mixture of Fe^2+^ and Fe^3+^ ions. The selected samples of systems I and III form FCC cells with a cell parameter *a* = 10.220(9) Å for sample M1, *a* = 10.267(1)Å for sample M11 (Table 2), whereas samples from system II form triclinic cells (Table 2) with *a* = 10.056(2) Å, *b* = 10.078(2) Å, *c* = 10.070(3) Å, α = 89.96(9)°, β = 88.47(2)°, γ = 90.39(9)° for sample M6. Some impurities were detected in sample M11, whose origin could not be assigned; however, the identified peaks corresponded to the FCC phase. Samples M1 and **M11** depict a spatial group *Fm-3m,* whereas sample M6 presents a *P1* spatial group and is consistent with reports from others [40,41]. Crystallite sizes obtained by the Scherrer equation are as follows: M1 = 193.88 Å, M6 = 295.72 Å, and M11 = 307.18 Å. These differences may be attributed to the intrinsic nucleation and growth rates in each CoHCF. Further experimentation should be carried out to prove this surmise. To better appreciate the relationship of metal concentrations during the synthesis procedure on the final crystalline phase of samples, two samples are selected for the analysis, namely M1 and M11. If the 200-lattice plane is magnified from sample M11 (Figure 4b), the diffraction peak is shifted to higher angle values, i.e., from 2θ = 17.26 to 2θ = 17.39 compared to sample M1 because more labile K^+^ ions replace less labile Na^+^ ions caused by the difference in their hydrated radii. These results are consistent with reports from the literature [42].

### 3.5. XPS Study

XPS was used to authenticate the presence of transition metals to confirm their oxidation state of the CoHCFs. Figure 5a shows the wide scan spectra for selected samples (M1, M6, and M11). The signals correspond to C 1s, O 1s, N 1s, Na 1s, K 2p, Fe 2p, and Co 2p; Figure 5b–h shows high-resolution core level spectra corresponding to these signals; Appendix A summarizes the results of deconvoluted peak assignments. The C 1s spectra of samples M1 and M6 are fitted with five peaks for sample M1 and four peaks for samples M6 and M11 (Figure 5b), respectively. The two peaks at 283.50 and 283.55 eV in samples M1 and M6 are assigned to the C-Fe^2+^ bond by Vannenberg [43] of the CoHCFs, while sample M11 shows two peaks located at 283.55 and 284.03 eV attributed to C-Fe^2+^ [43] and C-Fe^3+^ [43] bonds by Vannenberg [42]. This result indicates electron activity by charge transfer within sample M11. Peaks at 283.48, 284.53, 285.4, and 287.42 eV in sample M1 are assigned to the CO-Co [44] bond, C (elemental) [45], C from the environment [43] and to Co(CN)_x_, respectively. Sample M6 shows peaks at 284.47 and 285.60 eV corresponding to C (elemental) [45] and C from the environment [43], respectively. These last two assignments correspond to secondary species not removed and formed during the synthesis process and/or due to contact of the sample with the surroundings. Figure 5c shows the O 1s core level spectra. In sample M1, signals are fitted with two peaks, sample M6 needs four peaks, and sample M11 requires three peaks. The three samples present peaks at 535.4 eV [46], 532.8 eV [47], and 532.4 eV [48], all assigned to water molecules of samples M1, M6, and M11, respectively. In addition, sample M1 presents another peak at 531.4 eV corresponding to Co(NO_3_)_2_ [49]. Sample M6 shows peaks at 530 eV assigned to CoO [48] and at 531.3 eV corresponding to Co(NO_3_)_x_ [50]. Sample M11 presents two peaks, one at 530.91 eV assigned to Fe_x_O_y_ [49,51] and the other one at 531.88 eV attributed to Co(NO_3_)_x_ [52]. The core level spectra of N 1s are shown in Figure 5d. The signal for sample M1 requires three peaks, while samples M6 and M11 need four peaks each for fitting. Sample M1 depicts a peak at 396.53 eV assigned to the N-Co^2+^ bond [49], another peak at 396.95 eV corresponding to the N-Co^3+^ bond [43], and a third peak at 400.88 eV assigned to Co(NO_3_)_x_ [49]. In the case of sample M6, peaks are present at 397.30 eV of the N-Co^2+^ bond [43], at 397.76 eV assigned to the N-Co^3+^ bond [43] and small peaks at 401.79 and 406.7 eV corresponding to traces of Co(NO_3_)_x_ [49] and KNO_3_ [53], respectively. Fittings of sample M11 present peaks at 397.21 eV assigned to the N-Co^2+^ bond [43], another peak at 397.7 eV corresponding to the N-Co^3+^ bond [43], and a third peak at 396.55 eV assigned to a secondary compound of the type Co_x_N_y_ [54]. Figure 5e shows the Na 1s core level spectrum of sample M1. One peak at 1071.34 eV is required to fit the data and is attributed to Na of the compound Na_n_Co^2+, 3+^[Fe(CN)_6_] [43]. On the other hand, Figure 5f shows the K 2p core level spectra of samples M6 and M11, respectively. Both samples show two peaks at 292.41 and 292.62 eV, both assigned to K in the compound K_n_Co^2+,3+^[Fe(CN)_6_] [43]. Some satellites are present in these samples. Figure 5g shows the Fe 2p spectra resolved in individual peaks. Sample M1 requires one peak at 707.34 eV corresponding to the C-Fe^2+^ bond [55]; the rest of the peaks correspond to Auger-Co and satellites; sample M6 shows two peaks at 709.23 and 709.49 eV, both assigned to the C-Fe^2+^ bond [55] and another peak at 708.23 eV assigned to Fe° [56] overlapped with a satellite; the rest of the peaks correspond to Auger-Co and satellites. Sample M11 depicts three main peaks, one located at 712.72 eV corresponding to the C-Fe^2+^ bond [55], the second one at 715.27 eV assigned to the C-Fe^3+^ bond [55], and the third one at 711.7 eV assigned to Fe_x_O_y_ [57]. The XPS results for Fe 2p indicate that when CoHCFs are synthesized with Fe^2+^ ion in ferrocyanide, iron maintains a stable oxidation state. Conversely, when the Fe^3+^ ion in ferricyanide is used during the synthesis, iron presents electron activity by changing its oxidation state. Figure 5h shows the Co 2p core level spectra resolved in individual peaks. Sample M1 depicts two important peaks, the first at 780.68 eV corresponding to the N-Co^2+^ bond [49,58] and the second at 783.59 eV assigned to the N-Co^3+^ bond [13]. Sample M6 presents two relevant peaks, the first at 781.90 eV corresponding to the N-Co^2+^ bond [49,58] and the second at 783.58 eV assigned to the N-Co^3+^ bond [13]. Sample M11 shows two peaks at 785.54 and 786.36 eV assigned to N-Co^2+^ [49,58] and N-Co^3+^ [13], respectively. These results indicate that Co also presents spontaneous electron transfer reflected in changes to its oxidation state. The XPS results indicated that Fe^2+^, Co^2+^, and Co^3+^ were detected in samples M1 and M6 only, while in the case of sample M11, both Fe and Co presented the two oxidation states. The XPS technique was used to obtain the formulas for samples with the highest concentration of Co by separating the oxidation states of Fe and Co and the results obtained were: M1 = ${\mathrm{Na}}_{0.61}{\mathrm{Co}}_{0.95}^{\mathrm{II}}{\mathrm{Co}}_{0.5}^{\mathrm{III}}\left[{\mathrm{Fe}}^{\mathrm{II}}{\left(\mathrm{CN}\right)}_{6}\right]*8.6H_{2}O,$ M6 = $K_{1.52}{\mathrm{Co}}_{0.71}^{\mathrm{II}}{\mathrm{Co}}_{0.36}^{\mathrm{III}}\left[{\mathrm{Fe}}^{\mathrm{II}}{\left(\mathrm{CN}\right)}_{6}\right]*1.6H_{2}O$, M11 = $K_{0.12}{\mathrm{Co}}_{1}^{\mathrm{II}}{\mathrm{Co}}_{0.52}^{\mathrm{III}}\left[{\mathrm{Fe}}_{0.67}^{\mathrm{II}}{\mathrm{Fe}}_{0.33}^{\mathrm{III}}{\left(\mathrm{CN}\right)}_{6}\right]*6.6H_{2}O$. In conclusion, even with the large number of assignations ascribed to the different bonds within the CoHCFs, those with correspondence to two or more spectra of different elements are studied and provide information confirming the different oxidation states Fe and Co acquired during the synthesis procedure as well as the different bonds involved in the structure of the CoHCFs that ultimately will have an impact on the CO_2_ capture.

### 3.6. Surface Morphology of CoCHFs

The surface morphologies of the CoHCFs materials were examined by TEM (Figure 6) and by SEM (Figure 7) for samples M4 and M11, respectively. These samples were selected randomly for this study. It is demonstrated that the particles of CoHCF materials are typical nanocubes with particle sizes in the range of 80–130 nm (Figure 6a–d). For sample M11, aggregation of nanocubes is clearly observed (Figure 7a,b), suggesting that high molar ratios of K_3_Fe(CN)_6_:Co(NO_3_)_2_ lead to the aggregation of the precipitated CoHCF. However, from the magnification of SEM images, it can be seen that CoHCFs have rough surfaces.

### 3.7. CO_2_ Adsorption on CoHCFs

#### 3.7.1. Equilibrium Adsorption Isotherms

The results of CO_2_ adsorption equilibrium measurements at 273 K and 760 mmHg on three representative samples of CoHCFs (M1, M6, and M11) are depicted in Figure 8a. As it is expected from the results of N_2_ adsorption and XPS measurements, sample M11 shows the highest CO_2_ adsorption capacity (3.04 mmol g^−1^), followed by sample M1 (2.45 mmol g^−1^) and sample M6 (0.31 mmol g^−1^). Consequently, sample M11 is used to further perform all CO_2_ characterization experiments so that the diffusion processes of CO_2_ can be facilitated and the adsorption sites can be easily accessed by CO_2_ molecules. The high adsorption capacity presented by sample M11 is comparable with capacities reported for CoHCFs [59]. The CO_2_ adsorption and desorption isotherms at temperatures of 273–353 K on sample M11 are depicted in Figure 8b. The isotherms correspond to Type I b according to the IUPAC classification [30] and exhibit strong desorption hysteresis. These hysteresis results can be attributed to either of the following two reasons. On the one hand, Zhao et al. [60] report that the desorption hysteresis can be due to the diffusion of the gas through the intra-crystalline structure caused by the entanglement of CO_2_ molecules inside the vacancies of the CoHCFs upon desorption; further admission of CO_2_ to the material continues taking place at the onset of the desorption, and this is reflected by an increase in gas removal. On the other hand, Riascos-Rodriguez et al. [61] report that this type of hysteresis can be attributed to the expansion of the material structure, which appears when the filling of pores by the gas takes place when the material is in its relaxed state.

#### 3.7.2. Kinetic Measurements

To be a good adsorbent, the material must possess fast adsorption kinetics as one of the most important properties, according to Serna-Guerrero et al. [62]. This is relevant, especially in dynamic operations (e.g., in fixed bed adsorption separations) where the adsorbent ought to stand up high solute flows as related to its rate of adsorption [62]. As a result, a pivotal step in CO_2_ adsorption studies should include kinetic measurements. Figure 9a depicts the kinetic profiles at temperatures 273, 298, and 313 K and low absolute pressures. These results indicate the strong dependence of the adsorption rate on the temperature. For increased temperatures, CO_2_ uptakes are faster due to boosted kinetics, but the adsorption capacities diminish due to thermodynamic constraints [63]. This is consistent with the exothermal condition of the adsorption process. Conversely, the equilibrium is achieved at shorter times and higher temperatures, as can be observed by the sharper curves, and this is in line with rapidly increased diffusion and mass transfer phenomena. In general, very fast kinetics can be observed to occur within the first 10 s of CO_2_ capture at 273 K, whereas between ~60 and ~80% of CO_2_ is adsorbed within the first 6 s in the case of 298 K and is much faster in the case of 313 K, to within 3 s of contact time.

Using the Arrhenius equation [62], the activation energy (*E_a_*) of CO_2_ adsorption on CoHCFs is determined employing the mass transfer coefficients determined from the pseudo-second-order model at the three different temperatures studied. The Arrhenius equation is given by:(2)$$k_{f}=Aexp\left(-\frac{E_{a}}{RT}\right)$$
where: *k_f_* (g mmol^−1^ s^−1^) is the mass-transfer constant, *A* (g mmol^−1^ s^−1^) is the Arrhenius factor, *T* is the adsorption temperature (K), *R* (8.314 J mol^−1^ K^−1^) is the universal gas constant, *E_a_* (J mol^−1^) is the activation energy of adsorption. Parameters *k_f_* and E*_a_* are determined from the slope and intercept of a plot of ln *k_f_* vs. 1/T, respectively.

An activation energy E*_a_* = 35.01 kJ mol^−1^ and an Arrhenius factor *A* = 31.673 × 10^6^ g mmol^−1^ s^−1^, Figure 9b. A positive value of E*_a_* indicates that the adsorption rate increases as the temperature increases. This value is within the range of a physical CO_2_ adsorption process on CoHCFs (<40 kJ mol^−1^) [64].

#### 3.7.3. Effect of Adsorption Temperature on CO_2_ Uptake

One of the operating conditions that can significantly affect the magnitude of CO_2_ uptake by CoHCFs is the temperature. The results of the comparative study of CO_2_ removal by CoHCFs (sample M11) are depicted in Figure 10a. An adsorption capacity of 2.72 mmol g^−1^ is observed at 273 K, which is approximately 4.4 times higher than the one corresponding to 353 K (0.62 mmol g^−1^). The CO_2_ adsorption capacities decrease from 2.72 to 0.62 mmol g^−1^ as the temperature increases. This can be explained by the fact that the interactions between the cyano groups of CoHCFs and CO_2_ molecules are exothermic; the strong observed temperature dependence corresponds to a predominant kinetic diffusion-controlled process rather than a thermodynamic one. The increase in temperature prevents CO_2_ molecules from being transferred from the bulk to the surface of CoHCFs dictated by the kinetic hindrance. This, in turn, causes a diminution of the number of available adsorption sites at elevated temperatures.

#### 3.7.4. Isosteric Heats of Adsorption (Q_st_) of CO_2_ on CoHCFs

The heat released upon an adsorption process at certain coverage is defined as the isosteric heat of adsorption, Q_st_ [10,60]. This is a thermodynamic property indicating the strength of bonding between the molecules of a gas and the surface of a solid [10]. It can be used to reflect the affinities of guest gas molecules to the adsorbent. In general, a high value of Q_st_ is advantageous at low pressures, but a very large value of Q_st_ produces an elevated energy demand for regeneration purposes of the solid adsorbent [65]; a value of Q_st_ neither too large nor too small is suitable for CO_2_ removal at low pressures. Plots of ln P_CO2_ vs. 1/T shown in Figure 10b at constant coverage (called isosteres) in the range of 0.10–0.62 mmol g^−1^ and in the temperature range of 273–353 K were used to assess Q_st_ from the slopes. The isosteres present negative slopes being concordant with an exothermic CO_2_ adsorption process. It has been reported [66] that by monitoring the magnitude and changes of Q_st_ throughout the CO_2_ uptake process, it is possible to obtain information regarding the interactions at the molecular level involving CO_2_ molecules and the adsorbent and to elucidate the energetic diversity of the adsorbent surface [66]. Figure 10c shows the results of the calculated values of Q_st_ as a function of the amount adsorbed. We can observe a range of Q_st_ of 24.07–25.80 kJ mol^−1^ and an average value of 24.34 kJ mol^−1^, which falls within the range of physical adsorption [67]. The trend observed in Q_st_ with the amount adsorbed is that it first decreases to a minimum, then slightly increases, and finally decreases again. This is attributed to a high degree of surface heterogeneity and changes in the sorbate/sorbent and sorbate/sorbate interplays [68]. The main adsorption sites involved in the CO_2_ uptake include K and Na ions since these have basic characteristics to bind acidic CO_2_ molecules. In contrast, Co and Fe atoms are classified as Lewis acids that do not bind CO_2_ molecules easily due to the acidic nature of this gas. It was reported very recently [24,59] that the K^+^ ions exposed after the removal of water molecules from the crystallites of MOFs and successfully implanted in the MOFs produce adsorption sites with a strong affinity towards CO_2_ molecules through a symbiotic effect, and the K^+^ ions act effectively as gas confinement sites improving the CO_2_-framework affinity, thus enhancing the values of Q_st_. Q_st_ values show a decreasing trend as CO_2_ surface covering increases indicating that the CoHCFs are distinguished by a high degree of heterogeneity and showing differences in the adsorbate–adsorbent, and adsorbate–adsorbate interplays. Generally, different classes of adsorptive molecular forces (e.g., Van der Waals, hydrophobicity, or chemical bonds) correlate Q_st_ to the interplays between CO_2_ molecules and adsorbent surface [69]. The trends of Q_st_ observed in Figure 10c indicate that at the initial adsorption phases, a substantial amount of unoccupied adsorption sites is exposed on the surface of CoHCFs, and more molecules of CO_2_ can contact the surface of the adsorbent and the strength of adsorptive forces between CO_2_ molecules and the adsorbent is large, thus resulting in a low activation energy and high Q_st_. As the CO_2_ uptake process continues, an increased amount of occupied adsorption sites is present, and the adsorption of more CO_2_ molecules is burdensome [70]. In addition to this, adsorbate–adsorbate interactions start to take place, and attraction–repulsion London dispersion forces start acting between already adsorbed CO_2_ molecules [68]. Consequently, when the removal of CO_2_ continues, activation energy soars, and Q_st_ declines [70].

#### 3.7.5. Regenerability

The stability of adsorbent, after adsorption/desorption cycles and uptake capacity, is important as part of the properties of CoHCFs if they are to be used in industrial applications for CO_2_ capture [10]. In this study, adsorption/desorption cycle tests on sample M11 of CoHCFs were conducted at 298 K, and the results are shown in Figure 10d. The adsorption remains at ca 1.77 mmol g^−1^ during the first two cycles (100 and 99.9% adsorption efficiency). In the third cycle, the adsorption decreases to 1.74 mmol g^−1^ (98.4%), representing only ~1.6% loss. This marginal loss can be attributed to the possible blocking of pores by CO_2_ molecules to the adsorption sites and which can be removed along with water molecules as the repeated adsorption/desorption cycles were performed, whereas adsorption capacity increased to just 1.83 mmol g^−1^ in the fourth and fifth cycles, respectively. This slight increase in adsorption capacities in the last two cycles can be attributed to the further removal of water molecules from the CoHCFs lattices that possibly prevailed trapped in the interconnected channels of the materials during the conditioning process. These results are indicative that chemical functionalities and textural properties (i.e., S_BET_ and V_t_) remain unchanged and that the CoHCFs are reversible adsorbents for CO_2_ capture.

#### 3.7.6. Selectivity

The selectivity of CoHCFs for a mixture of gases was studied in this work. Generally, flue-discharged gases in industrial or power plants contain ~15% CO_2_ with mostly balanced N_2_. Consequently, measurements were conducted at 298 K, at a pressure range of 1–760 mmHg, and CO_2_:N_2_ = 15:85 using pure component adsorption isotherms of CO_2_ and N_2_ to mimic industrial flue gases. The CO_2_ and N_2_ gases are assumed as ideal gases, and their adsorptive properties are taken individually [71]. Selectivities are calculated using the Ideal Adsorption Solution Theory (IAST) [72]. This method uses the following Equation to calculate selectivities [73]:(3)SCO2/N2=qCO2qN2pCO2pN2
where S_CO_2_/N_2__ is the selectivity of CO_2_ over N_2_, q_CO_2__ and q_N_2__ represent the amounts of CO_2_ and N_2_ adsorbed at the absolute pressures of p_CO_2__ and p_N_2__, respectively.

Figure 11a compares the two adsorption isotherms on sample M11 of CoHCF at 298 K. It shows a significant difference in the amount of adsorbed CO_2_ compared to that of N_2_. At 760 mmHg, the amounts of CO_2_ and N_2_ adsorbed are 1.79 mmol g^−1^ and 0.10 mmol g^−1^, respectively. This is a significant difference reflected in the selectivity of CO_2_ over N_2_. Figure 11b depicts selectivity calculation results and reveals that a maximum selectivity of ~47 is reached at a pressure of 30.7 mmHg, and it starts to slowly decrease after 35 mmHg until a minimum of 16.7 is attained at 760 mmHg. The highly enhanced selectivity of CO_2_ over N_2_ has been attributed to small pore sizes in CoHCFs and PBAs and powerful dipole–quadrupole interactions acting in the extremely polarizing exposed metal sites cladding the cavities, which are able to distinguish between these two gases. Since CoHCFs are capable of selectively separating gases (CO_2_ over N_2_) from a mixture of gases, they can be potentially used in flue gas industrial applications.

## 4. Conclusions

Three K or Na-CoHCFs systems were synthesized with different stoichiometries by the co-precipitation method at mild reaction conditions to establish which stoichiometry is more efficient in the removal of CO_2_. The stoichiometry of samples from system III produced the best material for CO_2_ uptake. The three proposed systems were studied through a battery of analytical techniques, including N_2_ adsorption, TGA, FTIR, XRD, and XPS. Based on the results of N_2_ adsorption measurements, this work revealed the effect that alkali ions such as K^+^ or Na^+^ have on the final porous structure of the CoHCFs. It was found that samples containing K^+^ ions, a mixture of Fe^2+/^Fe^3+^ ions, and a higher concentration of Co^2+^/Co^3+^ ions in the CoHCFs structure in system III exhibited a porous structure with a much more developed external specific surface area, S_ext_ (72.69–172.18 m^2^ g^−1^) compared with samples containing Na^+^ and Fe^2+^ ions (systems I and II). The specific surface areas decreased as the Na_4_[Fe(CN)_6_]/Co(NO_3_)_2_ and K_4_[Fe(CN)_6_]/Co(NO_3_)_2_ molar ratios decreased for samples of systems I and III but not for samples of system II, and it was attributed to the fact that high molar ratios of Na_4_[Fe(CN)_6_]/Co(NO_3_)_2_ and K_4_[Fe(CN)_6_]/Co(NO_3_)_2_ favored the formation of CoHCFs cubic cells with high BET surface area. The results from N_2_ adsorption measurements provided evidence of the mesoporosity gained during the synthesis, along with the formation of interconnected micro tunnels and cavities formed in the crystalline structure of CoHCFs. TGA study revealed that the oxidation states of Fe, Co, and the types of alkali ions affected the number of water molecules contained in the samples and, consequently, the final porous structure of the material. XRD results allowed us to determine the formation of two different crystalline cells, FCC for systems I and III and triclinic for system II. These differences were attributed to the fact that samples in system I have Na^+^ and Fe^2+^ ions, system II samples contain K^+^ and Fe^2+^ ions, and system III samples have K^+^ and a mixture of Fe^2+^ and Fe^3+^ ions. XPS studies confirmed the different oxidation states of Fe and Co obtained during the synthesis of CoHCFs and allowed us to authenticate the charges involved in the stabilization of some of the CoHCFs structures. The CO_2_ adsorption isotherms indicated that the CoHCFs provided removal capacities for CO_2_ with a maximum CO_2_ removal capacity of 3.04 mmol g^−1^, comparable with other sorbents presented in the literature. The low calculated isosteric heat of adsorption of ca. 24.3 kJ mol^−1^ provides support that these materials can be used in batch and continuous CO_2_ removal operations due to low amounts of heat released that can make the adsorption process more controllable. The values of isosteric heats observed are attributed to the presence of K^+^ ions exposed after the removal of water molecules from the crystallites of K-CoHCFs, creating adsorption sites with a strong affinity towards CO_2_ molecules, and the K^+^ ions act as gas confinement sites improving the CO_2_ affinity for the CoHCFs, thus enhancing the values of Q_st_. The CoHCFs showed significantly fast CO_2_ adsorption kinetics (3 to 6 s to reach equilibrium), a property that makes them attractive for continuous adsorption operations. CoHCFs showed excellent efficiency towards adsorption/desorption cycles with a loss of ~1.6% of adsorption capacity. CoHCFs presented excellent selectivity for gas mixtures with selectivity factors of ca. 47 of CO_2_ over N_2_, thus demonstrating the potential application in flue gas operations. The limitations of this work include the CO_2_ removal efficiencies presented by the shapes of the porous structure and particle clustering composing the structures of the CoHCFS, which is, in turn, reflected in diffusional restrictions. As a possible prospect to help overcome these limitations, the synthesis of new CoHCFs is proposed using other synthesis routes, such as inverse microemulsion techniques that would help disperse the CoHCFs nanoparticles that are expected to reduce diffusional resistances during the CO_2_ uptake process.

## Figures and Tables

**Figure 1 materials-16-00608-f001:**
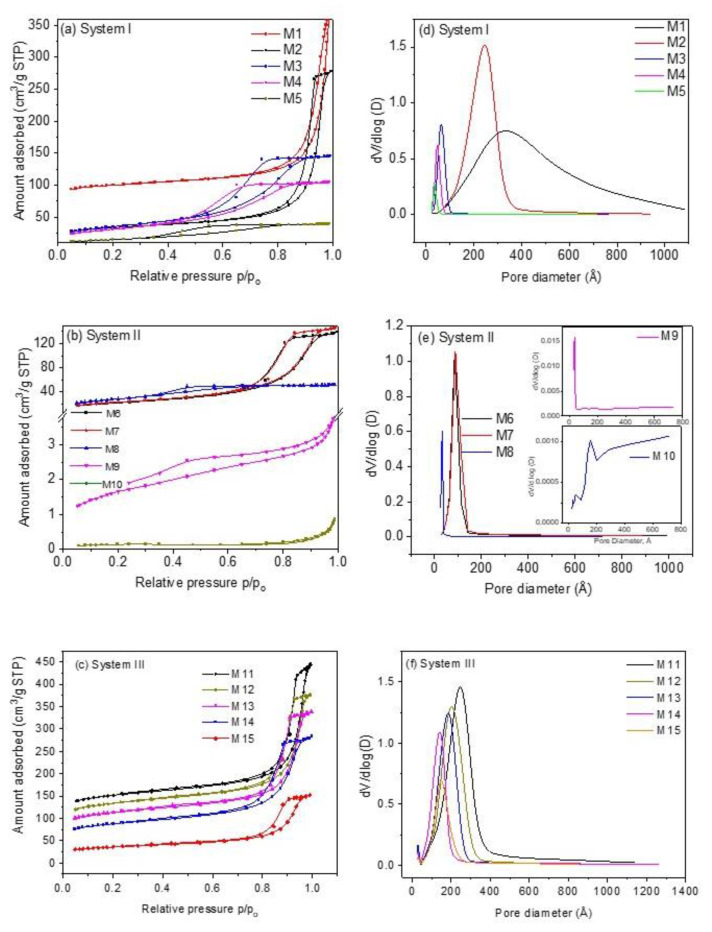
Nitrogen adsorption and desorption isotherms of CoHCFs of (**a**) system I of samples M1–M5; (**b**) system II of samples M6–M10; (**c**) system III of samples M11–M15. Pore size distributions of CoHCFs of (**d**) system I of samples M1–M5; (**e**) system II of samples M6-M-10; (**f**) system III of samples M11–M15.

**Figure 2 materials-16-00608-f002:**
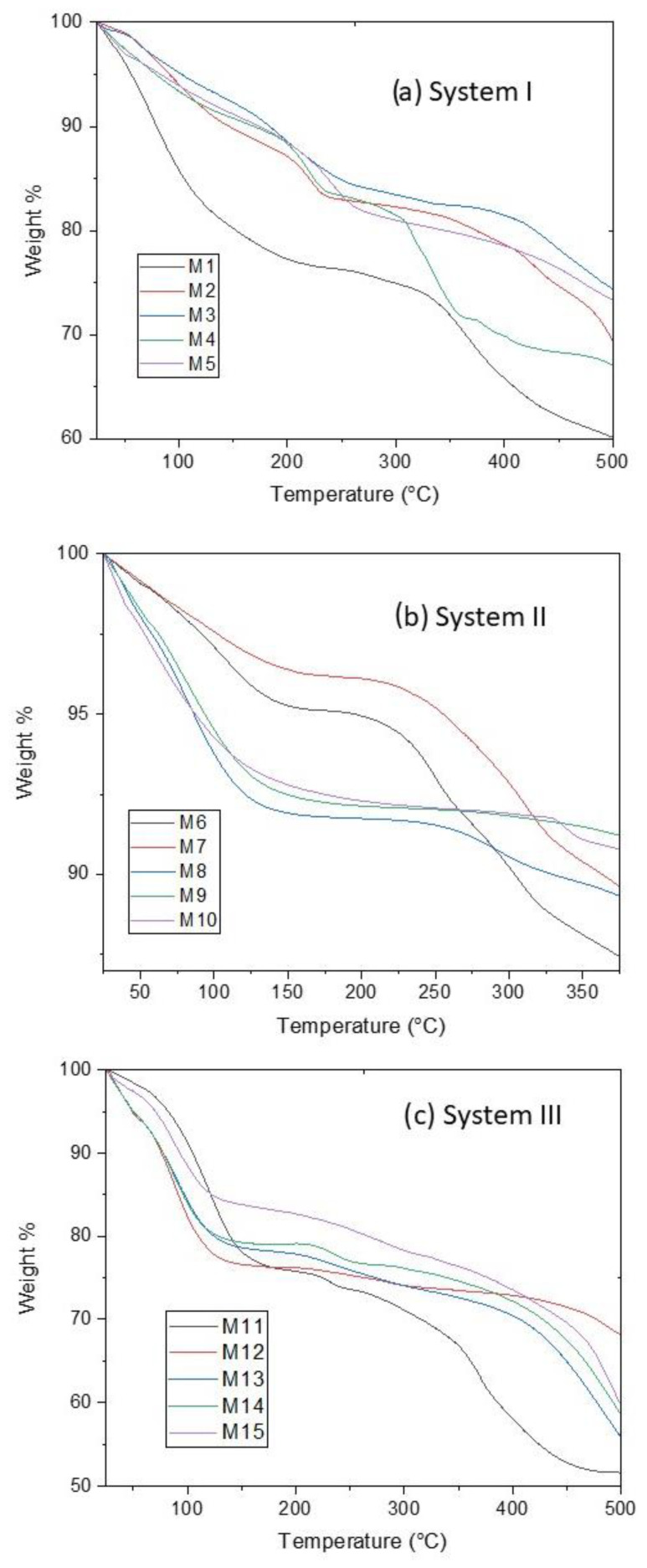
TGA results on CoHCFs for (**a**) system I; (**b**) system II; (**c**) system III.

**Figure 3 materials-16-00608-f003:**
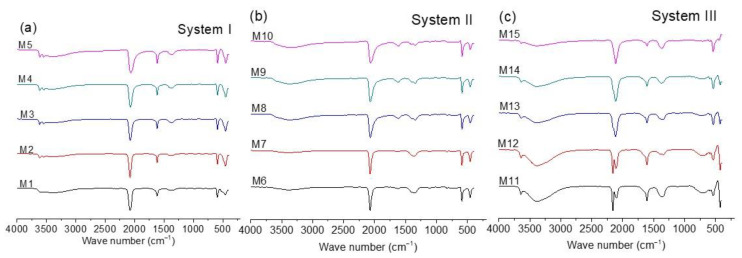
FTIR spectra for CoHCFs prepared by mixing Co(NO_3_)_2_ with (**a**) Na_4_[Fe(CN)_6_], system I; (**b**) K_4_[Fe(CN)_6_], system II; (**c**) K_3_[Fe(CN)_6_], system III.

**Figure 4 materials-16-00608-f004:**
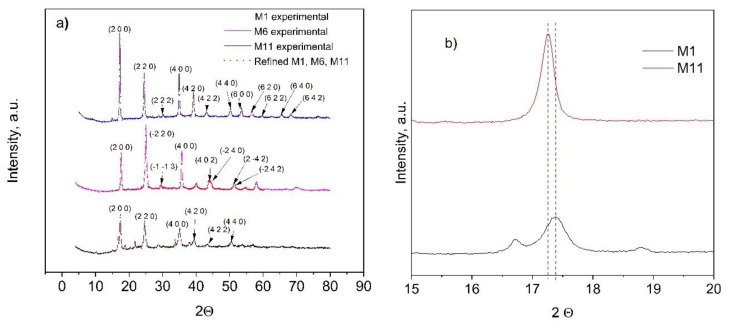
(**a**) Experimental (dots) and refined (continuous lines) powder X-ray diffraction data results for CoHCFs M1, M6, and M11; (**b**) partial magnification of the 200-lattice plane of M1 and M11 samples.

**Figure 5 materials-16-00608-f005:**
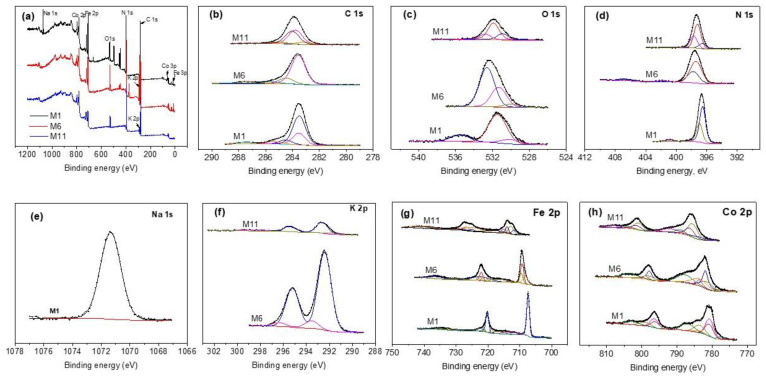
XPS core-level spectra on CoHCFs M1, M6, M11 (**a**) survey spectra; (**b**) C 1s; (**c**) O 1s; (**d**) N 1s; (**e**) Na 1s; (**f**) K 2p; (**g**) Fe 2p; (**h**) Co 2p.

**Figure 6 materials-16-00608-f006:**
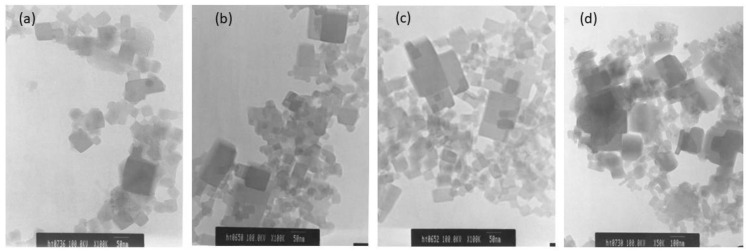
TEM micrographs of CoHCFs of sample M4 keeping fixed the molar ratio of Na_4_Fe(CN)_6_: Co(NO_3_)_2_ at 1.71:1, but using different amounts of reactants (**a**) 10 % wt, (**b**) 15 %wt, (**c**) 20 %wt, (**d**) 30 %wt.

**Figure 7 materials-16-00608-f007:**
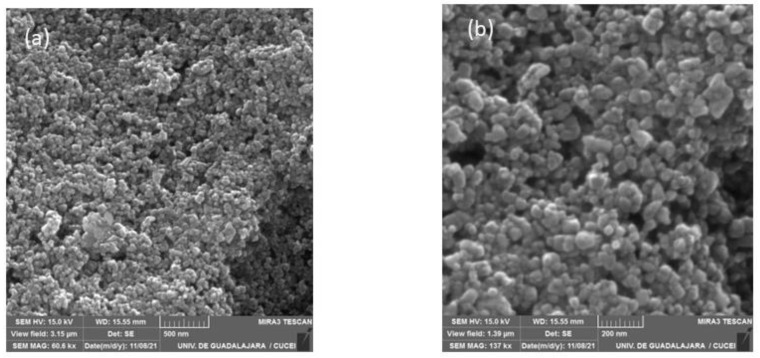
SEM micrographs of sample M11 at different magnifications (**a**) 500 nm and (**b**) 200 nm.

**Figure 8 materials-16-00608-f008:**
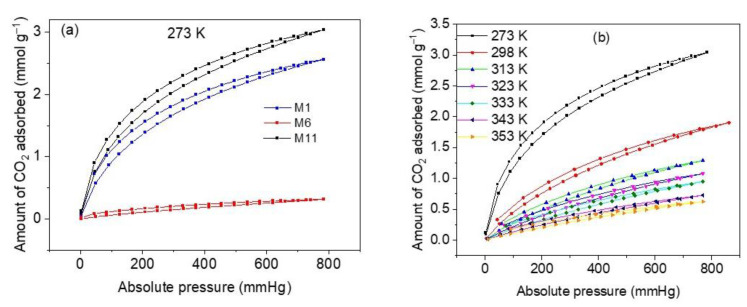
(**a**) CO_2_ adsorption capacities on samples M1, M6 and M11 of CoHCFs at 273 K; (**b**) CO_2_ adsorption isotherms at 273–353 K, on sample M11 of CoHCFs.

**Figure 9 materials-16-00608-f009:**
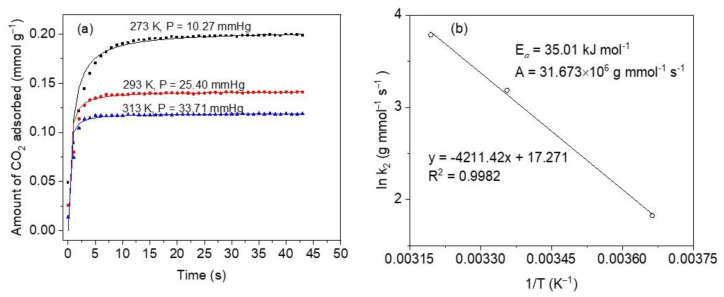
(**a**) CO_2_ adsorption kinetics at 273, 298 and 313 K, on sample M11 of CoHCFs; symbols represent experimental data and lines are for eye guidance; pressures up to 33.71 mmHg (4.5 kPa); (**b**) Arrhenius plot obtained from pseudo-second-order model constants.

**Figure 10 materials-16-00608-f010:**
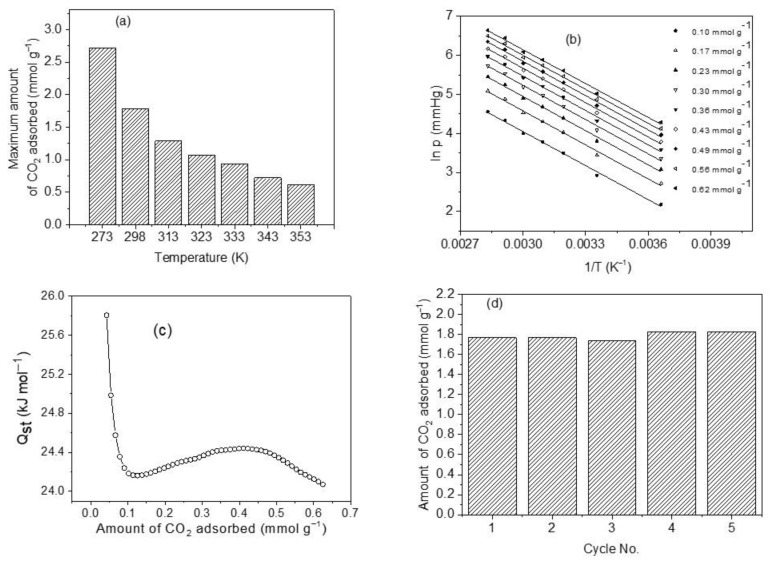
(**a**) CO_2_ adsorption capacity of sample M11 of CoHCFs at a temperature range of 273–353 K; (**b**) isosteres of CO_2_ adsorption as a function of temperature and amount adsorbed on sample M11 of system III of CoHCF; (**c**) change of isosteric heat of adsorption with the amount adsorbed for sample M11 of system III of CoHCFs using CO_2_ adsorption from 0 to 101 kPa; (**d**) recovery of sample M11 of CoHCFs after five cycles of CO_2_ adsorption/desorption.

**Figure 11 materials-16-00608-f011:**
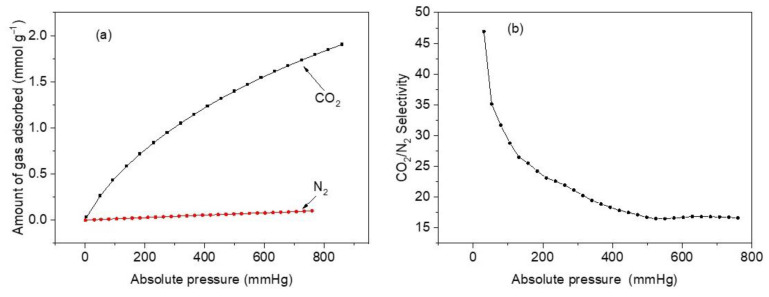
(**a**) CO_2_ and N_2_ adsorption isotherms at 25 °C and pressure between 1 and 760 mmHg for sample M11 of CoHCFs; (**b**) adsorption selectivity curve of CO_2_/N_2_ at the adsorption temperature of 25 °C, pressure range of 1–760 mmHg, and CO_2_:N_2_ = 15:85 for sample M11 of CoHCFs.

**Table 1 materials-16-00608-t001:** Textural properties of CoHCFs samples obtained from N_2_ adsorption measurements and stoichiometric coefficients for each sample.

Sample ID	Reactant Molar Ratio	S_BET_ (m^2^ g^−1^)	D_p_ (Å)	V_t_ (cm^3^ g^−1^)	S_ext_ (m^2^ g^−1^)	S_mic_ (m^2^ g^−1^)	V_mic_ (cm^3^ g^−1^)	V_meso_ (cm^3^ g^−1^)	Isotherm	Hysteresis	Formula
	Na_4_Fe(CN)_6_: Co(NO_3_)_2_	**System I**									
M1	1.71:2.5	311.91	64.25	0.444	78.80	233.11	0.1205	0.3235	IVa	H1	$Na_{0.61}Co_{1.45}\left[Fe{\left(CN\right)}_{6}\right]*8.6H_{2}O$
M2	1.71:2	110.83	155.23	0.417	73.06	37.77	0.0200	0.3970	IVa	H1	$Na_{0.97}Co_{1.29}\left[Fe{\left(CN\right)}_{6}\right]*4.9H_{2}O$
M3	1.71:1.5	122.43	73.48	0.224	105.88	16.54	0.0084	0.2156	IVa	H1	$Na_{1.83}Co_{0.92}\left[Fe{\left(CN\right)}_{6}\right]*4.2H_{2}O$
M4	1.71:1	109.53	59.12	0.162	97.99	11.54	0.0054	0.1566	IVa	H1	$Na_{2.01}Co_{0.88}\left[Fe{\left(CN\right)}_{6}\right]*4.4H_{2}O$
M5	1.71:0.5	52.78	46.34	0.061	49.15	3.63	0.0014	0.0596	IVa	H2b	$Na_{2.14}Co_{0.89}\left[Fe{\left(CN\right)}_{6}\right]*5.3H_{2}O$
	K_4_Fe(CN)_6_: Co(NO_3_)_2_	**System II**									
M6	1.71:2.5	77.63	109.09	0.216	68.13	9.50	0.0044	0.2116	IVa	H1	$K_{1.52}Co_{1.07}\left[Fe{\left(CN\right)}_{6}\right]*1.6H_{2}O$
M7	1.71:2	81.28	111.04	0.228	68.89	11.38	0.0053	0.2227	IVa	H1	$K_{1.88}Co_{0.90}\left[Fe{\left(CN\right)}_{6}\right]*1.1H_{2}O$
M8	1.71:1.5	101.41	31.67	0.085	91.65	9.74	0.0042	0.0808	IVa	H1	$K_{2.13}Co_{0.81}\left[Fe{\left(CN\right)}_{6}\right]*2.1H_{2}O$
M9	1.71:1	5.72	41.40	0.006	5.72	----	0.000074	0.0059	II	H3	$K_{2.4}Co_{0.71}\left[Fe{\left(CN\right)}_{6}\right]*1.9H_{2}O$
M10	1.71:0.5	0.45	106.69	0.001	0.4720	----	0.000012	0.0010	II	H3	$K_{2.27}Co_{0.79}\left[Fe{\left(CN\right)}_{6}\right]*2.2H_{2}O$
	K_3_Fe(CN)_6_: Co(NO_3_)_2_	**System III**									
M11	1.71:2.5	479.21	55.56	0.505	170.91	308.30	0.1603	0.3447	IVa	H1	$K_{0.12}Co_{1.52}\left[Fe{\left(CN\right)}_{6}\right]*6.6H_{2}O$
M12	1.71:2	425.83	54.69	0.452	172.18	253.65	0.1320	0.3200	IVa	H1	$K_{1.03}Co_{1.18}\left[Fe{\left(CN\right)}_{6}\right]*5.7H_{2}O$
M13	1.71:1.5	362.87	57.74	0.399	159.68	203.18	0.1059	0.2931	IVa	H1	$K_{1.49}Co_{1.04}\left[Fe{\left(CN\right)}_{6}\right]*5H_{2}O$
M14	1.71:1	283.70	60.94	0.374	146.18	137.51	0.0720	0.3020	IVa	H1	$K_{1.91}Co_{0.89}\left[Fe{\left(CN\right)}_{6}\right]*4.4H_{2}O$
M15	1.71:0.5	117.99	79.73	0.215	72.69	45.29	0.0238	0.1912	IVa	H1	$K_{2.21}Co_{0.78}\left[Fe{\left(CN\right)}_{6}\right]*3.9H_{2}O$

**Table 2 materials-16-00608-t002:** Crystallographic data and structure refinement parameters were obtained after the refinement of powder X-ray diffraction measurements on CoCHFs M1, M6, and M11.

Parameter	M1	M6	M11
Space group	Fm-3m	P1	Fm-3m
Crystal system	FCC	Triclinic	FCC
*a* (Å)	10.220(9)	10.056(2)	10.267(1)
*b* (Å)	10.220(9)	10.078(2)	10.267(1)
*c* (Å)	10.220(9)	10.070(3)	10.267(1)
α (deg)	90	89.96(9)	90
β (deg)	90	88.47(2)	90
γ (deg)	90	90.39(9)	90
Crystal size (Å)	193.88	295.72	307.18
Atoms per unit cell	Na_8_Co_4_Fe_4_C_24_N_24_	K_8_Co_4_Fe_4_C_24_N_24_	K_4_Co_4_Fe_4_C_24_N_24_

## Data Availability

Not applicable.

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
