# Peer review of "Porous Structural Properties of K or Na-Co Hexacyanoferrates as Efficient Materials for CO_2_ Capture"

_materials, 2023, doi:10.3390/ma16020608_

Round 1
Reviewer 1 Report
The manuscript “Porous structural properties of K or Na-Co hexacyanoferrates as efficient materials for CO2 capture by Paloma M. Frías-Ureña et al. the authors reports on the synthesis of three series of CoHCFs. Although the materials have been extensively characterized ( XRD, XPS, FTIR, N2 CO2 adsorption and kinetics..) their order of description and positioning in the manuscript make the readability and flow of the text difficult. The majors problems are the two points below (TGA interpretation and XRD Rietveld + phases) . The recommendation is to reconsider and rewrite the manuscript
Since in the synthesis you are using Co(NO3)2 how and why the presence of nitro groups is eliminated, especially in the TGA depiction ? Is it possible to have a NO3 in the M coordination and as leaving group? Besides in the XPS you are detecting such “signals”.
The XRD data for the triclinic seems odd. The top and middle diffractograms look identical. A sample displacement during the preparation may have produced the shift. The triclinic parameters are close to 90 degree and a=b=c=10.05. Please check the provided Atoms per unit cell as Na4CoFe(CN)6, K2CoFe(CN)6, K1CoFe(CN)6 as those do not Mach the XPS obtained formulas .
Could you provide in the SI the Rietveld refinement parameters and the atomic coordinates especially for the triclinic phase?
Minor
In the SI the TGA plots are given five per page. Please consider enlarging those.
L51 Please correct to superscript g-1
L79 The authors probably mean that two types of water positions can be discerned. Not two chemically distinct water molecules.
L81 the eighths of the crystal cell??? Is it at 0,0,0 a corner atom at the unit cell, a special position …please clarify?
The TGA data is supposedly up to 500 deg but in the SI figures the attained temperatures are 600 deg. Please check.
In the references a lot of CO2 and typing errors are visible. Please correct
Author Response
Response to Reviewer 1 Comments
Please see attachment

Reviewer 2 Report
Although CoHCFs have been extensively studied as good CO2 adsorbents, the relation between their porous structural properties and the efficiency of CO2 capture was not fully elucidated, as commented by the authors in introduction. This paper comprehensively reports the effect the synthesis stoichiometry on the porous structural properties of CoHCFs and on their efficient CO2 capture. The contents make a great degree of sense, and its quality is overall high. Therefore, I think this paper is publishable in Materials.
Nevertheless, the writing of the unit in figures should be unified before publication (e.g., compare Figs. 2, 3, and 8), mainly for improving its readability.
Additionally, (a) the symbol, "degree", after numbers in Fig. 8a is unnecessary, (b) the unit of the horizontal axis in Fig. 8b is lacking, (c) the number, “0.7”, in Fig. 8c is cut off, and (d) the word, “Amongst,” (L. 552, p.17) seems to be unnecessary.
Reviewer 3 Report
Highlight changes in yellow in a next revision, please. No track changes.
Specifically, in the case of the methods, similarity should be reduced.
Particularly considering that in many cases no references are present and the similarity is there all over.
Also in the cases of equations. All similarity needs to be addressed specifically because if there is known data then you need to include the necessary references immediately before citing the equations.
Please limit known data to the minimum.
Please address all italics in parameters too.
Similarity is also found in the results with entire sentences and no reference.
Authors should not use published sentences. Even if intended to say the same thing.
Again. Similarity can be found in the text in other cases and no references are cited whatsoever.
“using the Arrhenius equation,”
how is this considered results? When equations are presented in results, it should be the result obtained instead. These are introductory equations.
Again, please revise all italics in the manuscript.
General comments then:
I believe that an abstract, just like conclusions with similar structure and different content, should necessarily start by a brief contextualization, then brief methodology, main findings, and practical implications. In the case of conclusions, he limitations and future prospects should be added.
So it seems to me that the abstract should be revised in order to start contextualization this particular study and end clearly explaining why is it important.
I have made several comments before this. See that in the cases of figures the caption should be self-explanatory by its own, not forcing the reader to go back and see what is the difference between system one system. Use system three etc. The figure to me needs a better caption.
“9
Figure 1. Nitrogen adsorption and desorption isotherms of CoHCFs of: a) System I, b) System II, c) 300
System III. Pore size distributions of CoHCFs of d) System I, e) System II, f) System III”
The methods used to obtain these values do not need to be included in the caption, instead that is addressed in the text, so I would suggest working on every captions to be self-explanatory in terms of what is being shown.
“Table 1. Textural properties of CoHCFs samples obtained from N2 adsorption measurements and 302
stoichiometric coefficients for each sample”
The table should be better worked in terms of separating specific content and format. See the PDF.
Please consider avoiding abbreviations in captions as in headings, and again consider the previous comment as to the meaning of system one, system two, etcetera.
Compare the graphics in figure two with other graphics where units are not included inside curved brackets. So there is no coherence.
Why add the word results to results headings inside results section?
“3.6. Results of CO2 adsorption on CoHCFs”
miss address or other cases.
Of?
“3.6.5. Regenerability”
please avoid using the red color in bars. It is insulting in specific countries. Also, there is no need to add the percentage to each value. There are other ways to do this.
“Figure 8. a) CO2 adsorption capacity of sample M11 of CoHCFs at a temperature range of 273-353 624
K; b) Isosteres of CO2 adsorption as a function of temperature and amount adsorbed on sample M11 625
of system III of CoHCF; c) Change of isosteric heat of adsorption with the amount adsorbed for 626
sample M11 of system III of CoHCFs using CO2 adsorption from 0 to 101 kPa; d) Recovery of sample 627
M11 of CoHCFs after 5 cycles of CO2 adsorption/desorption”
again our Dez results or methodology?
“This method uses the following equa- 636
tion to calculate selectivities: 637
???2/?2 =
???2
⁄??2
???2
⁄??2
(3)”
again, authors need to start with a brief. Contextualization does clearly justify. Why is this manuscript being presented? Why is this study important?
“4. Conclusions”
Were or are? that the tenses are important.
In results, the present tense should be used.
“The values of isosteric heats observed 688
were attributed”
as it happened in the abstract. These are general superficial comments and with practical implications. Strong practical implications also add limitations and future prospects.
“thus 696
demonstrating the potential application in flue gas operations”
please check the journal guidelines and compare the references using upper letter with references using lower letter.
“16. T. KAWAMOTO, H. TANAKA, Y. HAKUTA, A. TAKAHASHI, D. PARAJULI, K. MINAMI, T. YASUTAKA, T. UCHIDA, Ra- 751
dioactive cesium decontamination technology for ash, Synth. English Ed. 9 (2016) 139–154. https://doi.org/10.5571/syn- 752
theng.9.3_139”
believe references would have to be entirely checked. For example, this one does not seem complete.
“71. A.L. Myers, J.M. Prausnitz, Thermodynamics of photosynthesis, A. I. Ch. E. J. 11 (1965) 121–127”
It is my perspective that supplementary material should not be referred in the manuscript. Inside the manuscript, because if it is supplementary, it can be used by readers only if they want to and if they look for it.
In figure one, for example, if it is a grouped figure, then the captions should necessarily address each sub caption by letter after the main caption, explaining in detail everything.
All units inside the graphics should be included in curved brackets as usual.
The graphics inside the supplementary materials have a very low quality, particularly considering that this is a word format, so they should all be revised.
The legends do not clarify the different lines used. That is, the line seems to be the same and one is M7 for example and another one is something else. So the readers will not understand what is being presented.
When references are addressed in a table, the names of the authors should be present before the reference number. It is then like this everywhere.
Again, all units should be inside curved brackets.
Please consider what is in fact essential to be presented under supplementary materials. All tables need to be revised in terms of format and aspect, it seems to me.
I would like to say that the comments are only intended to assist the authors in obtaining a better text. I understand and respect the experimental and extensive work involved in this manuscript, however. Some changes can be made to make it more relevant.
Round 2
Reviewer 3 Report
Highlight changes in yellow in a next revision, please. No track changes.
“Detailed” authors answers say mostly nothing…
They are also incorrect since in several cases changes were not really implemented
Similarity has been extensively extended, which is unacceptable.
Again:
“Particularly considering that in many cases no references are present and the similarity is there all over.”
Authors have just ignored it.
Again:
“Similarity is also found in the results with entire sentences and no reference.”
“Again. Similarity can be found in the text in other cases and no references are cited whatsoever.”
Authors do not clarify the presence of references in the results section (so why are the references there, please explain, since only the reference is unclear:
“peaks at 530 eV assigned to CoO [47] and at 531.3 eV corresponding to Co(NO3)x [49]. Sample M11 presents two peaks, one at 530.91 eV assigned to FexOy [48,50]
and many more…”
This says nothing, is this a comparison with the literature, what literature, one study, why? I theses cases, direct references with authors names are required.
Many of my comments were perfectly ignored and because of that, I suggest them to read them again.
in supplementary materials, the list of references needs to be included, of course, starting at 1, it is a separate document
